# Converting inorganic sulfur into degradable thermoplastics and adhesives by copolymerization with cyclic disulfides

Yuanxin Deng [1], Zhengtie Huang[1], Ben L. Feringa [1,2] ✉, He Tian [1], Qi Zhang [1] ✉ & Da-Hui Qu [1] ✉

Converting elementary sulfur into sulfur-rich polymers provides a sustainable strategy to replace fossil-fuel-based plastics. However, the low ring strain of eight-membered rings, i.e., $S_8$ monomers, compromises their ring-opening polymerization (ROP) due to lack of an enthalpic driving force and as a consequence, poly(sulfur) is inherently unstable. Here we report that copolymerization with cyclic disulfides, e.g., 1,2-dithiolanes, can enable a simple and energy-saving way to convert elementary sulfur into sulfur-rich thermoplastics. The key strategy is to combine two types of ROP—both mediated by disulfide bond exchange—to tackle the thermodynamic instability of poly(sulfur). Meanwhile, the readily modifiable sidechain of the cyclic disulfides provides chemical space to engineer the mechanical properties and dynamic functions over a large range, e.g., self-repairing ability and degradability. Thus, this simple and robust system is expected to be a starting point for the organic transformation of inorganic sulfur toward sulfur-rich functional and green plastics.

Sulfur is the fifth most abundant element on Earth and is found in millions of chemical products in modern industry. Cyclooctasulfur ($S_8$) is the most well-known allotrope of elementary sulfur and bears a chemical structure of an eight-membered ring consisting of eight sulfur atoms connected via disulfide bonds[1]. More than 60 million tons of sulfur $S_8$ are produced today[2]. Most of these are massive and valueless solid deposits because the niche downstream market for elementary sulfur is relative to its production scale. Hence, using elementary sulfur as the feedstock to produce value-added modern commodities, especially plastics, has been an increasingly attractive target in the past decade[3–7]. The representative application of elementary sulfur in the plastic industry is in vulcanization for crosslinking natural rubbers into tough elastomers[8]. This constitutes the main market and usage of elementary sulfur, but the scale is far below the amount of sulfur accumulated in modern industry.

One long-term challenge is converting elementary sulfur in applications as an alternative feedstock for producing synthetic plastics[9]. A representative strategy involves the use of inorganic sulfur as a sulfur source for the one-pot preparation of sulfur-containing polymers[10,11]. This highly attractive and fundamental approach, however, yields polymers with low sulfur content and relies on the use of organic solvents. Another more promising strategy for sulfur conversion is direct polymerization of elementary sulfur, i.e., using $S_8$ rings as cyclic monomers for ring-opening polymerization (ROP). However, the fundamental issue pertains to the inherent lack of ring strain of the cycloocta-sulfur ring, which provides a thermodynamically unfavorable starting point for ROP[12,13].

Previous studies have shown that plastic-like poly(sulfur) could be made by rapidly cooling molten sulfur[14,15], thus showing rubber-like properties, which are metastable at room temperature and, as a consequence, revert to cyclic $S_8$ monomers within hours to days. In 2013,

[1]Key Laboratory for Advanced Materials and Joint International Research Laboratory of Precision Chemistry and Molecular Engineering, Feringa Nobel Prize Scientist Joint Research Center, Institute of Fine Chemicals, Frontiers Science Center for Materiobiology and Dynamic Chemistry, School of Chemistry and Molecular Engineering, East China University of Science and Technology, Meilong Road 130, Shanghai 200237, China. [2]Stratingh Institute for Chemistry, Faculty of Science and Engineering, University of Groningen, Nijenborgh 4, 9747 AG Groningen, The Netherlands. ✉e-mail: b.l.feringa@rug.nl; q.zhang@ecust.edu.cn; dahui_qu@ecust.edu.cn

Chung et al. pioneered the technology called "inverse vulcanization", using dienes to crosslink poly($S_8$) polymers at 185 °C to produce sulfur-rich thermosetting materials[16]. The resulting material does not revert to $S_8$ monomers because of the kinetic trapping effect of the diene crosslinking. This milestone brings new opportunities for the valorization of elementary sulfur into sulfur-rich thermosets[17–25], which already offer a few promising applications, e.g., lithium–sulfur batteries[26–28], optical windows[29,30], and the removal of heavy metals[17,31,32]. The anionic ring-opening polymerization of $S_8$ was reported by Penczek et al. in 1978[9]. Very recently, Yang et al. discovered the anionic hybrid copolymerization of $S_8$ with acrylate[13]. This study introduced acrylates, thus enabling the random insertion of oligomeric poly($S_x$) into the backbone of poly(acrylate)s. Despite this notable progress in sulfur valorization, converting elementary sulfur into value-added thermoplastics with chemically tailorable structures and functions via an economically profitable methodology has remained elusive. Moreover, the valorization pathway of elementary sulfur remains to be diversified because all the state-of-the-art methodologies strongly rely on the specific sulfur-ene reaction to form robust C-S bonds. How to exploit another distinctive pathway to valorize elementary sulfur in an atomically efficient way that enables economic profitability, chemical diversity, and sustainability (e.g., self-repairing, degradability, etc.) has become a major challenge for the academic and industrial research community.

Here, we report that cyclic disulfides could copolymerize with elementary sulfur to produce high-performance and degradable thermoplastics and adhesives (Fig. 1). Owing to the strained disulfide

bonds of cyclic disulfides[33], the copolymerization reaction could be initiated at 120 °C without using any external catalyst. The robustness and sustainability of the reaction are underscored by a green synthesis that is free of solvents, inert gas protection, purification, and waste production. The reactions are promoted by the introduction of the ROP of five-membered cyclic disulfides to contribute to the ring-linear equilibrium from the aspects of both enthalpy and entropy. More importantly, the diverse sidechain groups of cyclic disulfide monomers offer significant chemical space to molecularly engineer the material properties of the resulting dry network[34]. These advantages give the sulfur-based polymers promising application prospects in diverse functional thermoplastic materials such as elastomers, self-healing materials, and degradable adhesives.

## Results

### Synthesis and characterization

Starting from a natural small molecule (thioctic acid (TA)), we synthesized a series of TA-based derivatives, including TAA, TABA, TADA, TAMe, and TAH (Fig. 2a). These were fully characterized by nuclear magnetic resonance (NMR) spectroscopy, high-resolution mass spectrometer (HR-MS), and Fourier transform infrared (FT-IR) (Supplementary Figs. 1–7). Copolymerization with elementary sulfur is straightforward: Simply mixing two types of powder samples at molten temperature (120 °C) and keeping the temperature for 2 h produced copolymer networks after cooling to room temperature (Fig. 2a and Supplementary Fig. 8). The resulting copolymers exhibited a yellow-to-

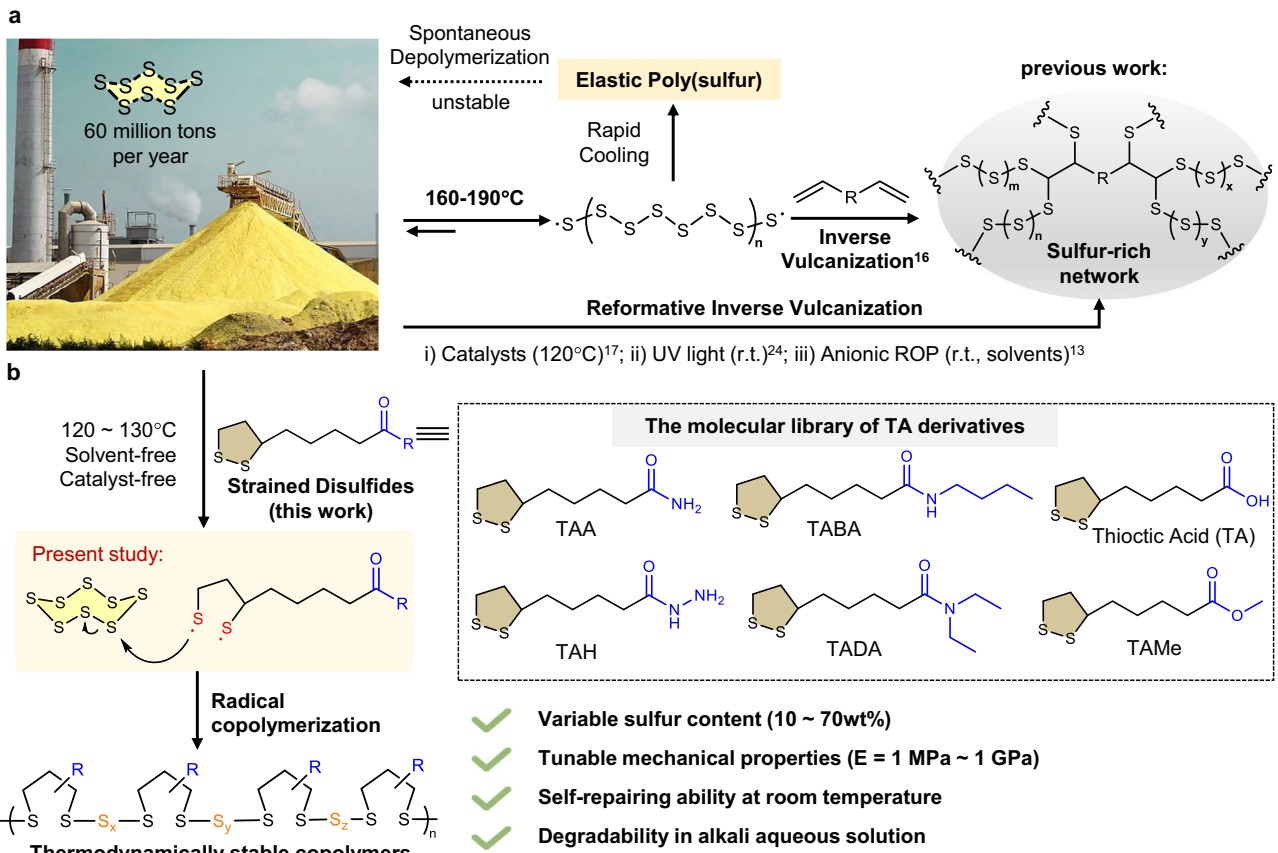

**Fig. 1 | Valorization strategies using elemental sulfur. a** One of the milestones of sulfur valorization was pioneered by Chung et al. in 2013 using the strategy called inverse vulcanization[16]. This is based on the efficient reaction between sulfur radicals and dienes. This method has been improved and updated by catalytic activation, photochemical reactions, and anionic copolymerization. UV light: ultraviolet light, ROP: ring-opening polymerization, r.t.: room temperature. **b** The concept of the present work, using strained disulfides to enable copolymerization of inorganic sulfur under greener conditions, offers another distinctive way to achieve high-value conversion of elementary sulfur.

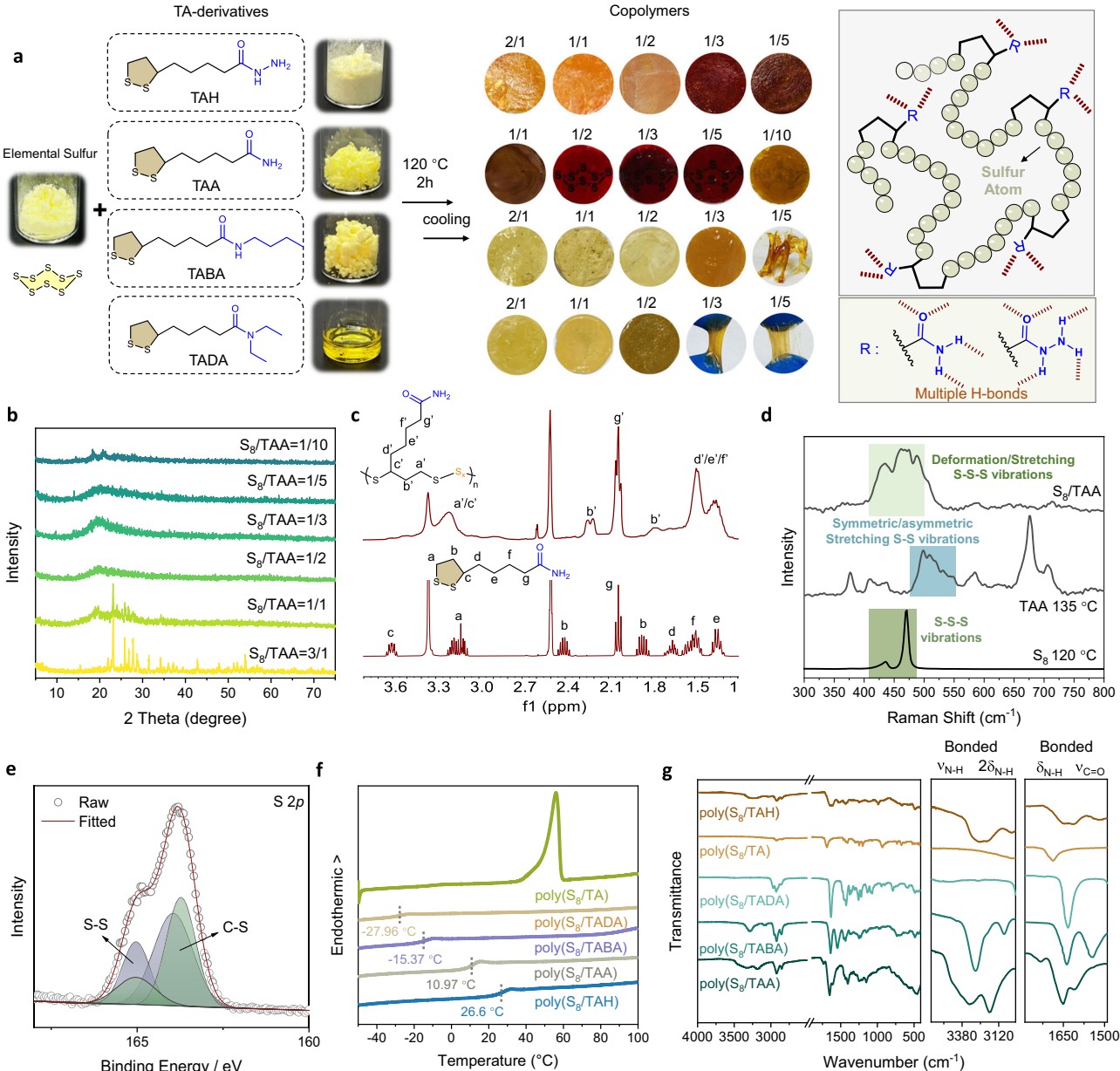

**Fig. 2 | Structure characterization of poly(S₈/TAA) copolymers. a** Preparation method of copolymerization and photographs of the resulting copolymers. **b** XRD pattern of the as-prepared poly(S₈/TAA) copolymer samples. **c** ¹H spectra of poly(S₈/TAA = 1/2) and TAA monomer in DMSO-$d_6$. **d** Partial Raman spectra of poly(S₈/TAA = 1/2), the product of TAA after heating, and the product of S₈ after heating. **e** S 2$p$ XPS spectrum and its deconvoluted peaks of S−S and C−S from poly(S₈/TAA = 1/2). **f** DSC of copolymers. **g** ATR-FT-IR spectra of the resulting copolymers with different monomers.

red-brownish translucent appearance depending on the sidechain of the disulfide monomers as well as the mixing ratio. The poly(S₈/TAMe) shows a viscous and flowing liquid behavior because of the lack of supramolecular sidechain crosslinking (Supplementary Fig. 9). The highest sulfur loading ratio could reach up to 70 wt%, while higher ratios of copolymers resulted in mixtures of sulfur crystals (excess S₈) and copolymers (Supplementary Fig. 10).

We first investigated the nature of the copolymer through several X-ray analyses and spectroscopic methods (Fig. 2b and Supplementary Figs. 11–15). X-ray diffraction (XRD) patterns indicated the amorphous nature of the resulting solid samples (Fig. 2b and Supplementary Fig. 11), while the samples prepared by melting each of the monomers (i.e., S₈ powders or TA-derivatives) were shown as semi-crystalline monomers/oligomers (Supplementary Figs. 12 and 13). Field emission scanning electron microscopy (FESEM) also indicates that the surface of the polymer film is smooth without any noticeable phase separation

or recrystallization of elemental sulfur (Supplementary Fig. 16). The characteristic proton absorptions of the 1,2-dithiolane ring in ¹H NMR spectra (Fig. 2c and Supplementary Fig. 17) further confirmed the open-chain structure in the resulting copolymers. While the vibration bands of carbonyl groups have been clearly observed in FT-IR spectra of the copolymers (Supplementary Fig. 7), the possible sulfuration reaction was further excluded by performing additional ¹³C NMR spectroscopic measurements of poly(S₈/TAA) and poly(S₈/TAMe) dissolved in DMSO-$d_6$, showing (i) the consistency of the carbonyl carbon peak (C$_h$ at 173 ppm) between monomers and polymers and (ii) the absence of sulfurated carbon peak (Supplementary Figs. 18 and 19). Furthermore, Raman analysis showed the characteristic vibration band of S−S−S and S−S bonds (i.e., 450–520 cm⁻¹; Fig. 2d and Supplementary Fig. 20), which can be considered as a signature of the successful ring-opening polymerization of elementary sulfur according to the previous literatures[12,13].

X-ray photoelectron spectroscopy (XPS) was also used to characterize the bonding environment of sulfur atoms in the network (Fig. 2e and Supplementary Figs. 21 and 22). The C 1s of poly($S_8$/TAA) was deconvoluted to C–C, C–S, C–O, and C=O at 284.6, 284.8, 285.3, and 288.2 eV, respectively (Supplementary Fig. 21). Deconvolution of the S 2p peak into two doublets led to C–S and S–S bonds at 163.7/165.0 and 164.0/165.1 eV (Fig. 2e)[12,21,35], showing a distinctive configuration from the polymer obtained from TAA via heating (Supplementary Fig. 22). Moreover, the solubility test in a series of organic solvents revealed the lack of solubility of the materials in most organic solvents except dimethyl sulfoxide (DMSO) and dimethyl formamide (DMF) (Supplementary Figs. 23). The analysis of matrix-assisted laser desorption/ionization time-of-flight mass spectrometry (MALDI-TOF-MS) further confirmed the direct copolymerization between cyclic disulfides and $S_x$ oligomers (Supplementary Figs. 24–26). Thermogravimetric analysis (TGA) showed the thermal decomposition temperature to be 180–240 °C for all the copolymer samples (Supplementary Fig. 27). Differential scanning calorimetry (DSC) analysis showed the glass transition temperature ($T_g$) of these copolymers finely tunable from −28 to 27 °C (Fig. 2f). The copolymers rich in H-bonds bear higher $T_g$ due to the additional noncovalent crosslinking. These structural characterization data confirmed the successful copolymerization of the TA-based derivatives and $S_8$ yielding amorphous random sulfur-rich copolymers.

To further evaluate the robustness of the radical copolymerization strategy, the polymerization reaction was also performed in the presence of a series of radical quenchers and scavengers that normally inhibit radical polymerization (Supplementary Fig. 28). Commonly used radical trapping agents, e.g., TEMPO, polymerization inhibitor (p-toluidine, chloranil, and hydroquinone), sulfur radical quencher (5-hexen-1-ol), have been used, and the resulting solids were characterized by XRD and Raman spectroscopy (Supplementary Fig. 29). XRD data revealed that the resulting copolymers with TEMPO, p-toluidine, or chloranil are amorphous. In the case of 5-hexen-1-ol or hydroquinone, a small amount of sulfur precipitation was observed. Raman spectra also suggest the occurrence of copolymerization (Supplementary Fig. 29). These results support the robustness of our system is impurity-tolerant, which should be attributed to the advantage of in-situ and continuous generation of sulfur radicals from cyclic disulfide monomers at the used temperature.

The amide sidechain of TAA and TAH monomers provides abundant H-bonds for the plasticization of the poly(sulfur) by forming a supramolecular crosslinked network. A series of FT-IR spectroscopic comparisons (Fig. 2g and Supplementary Figs. 30–34) of the copolymers prepared by different TA derivatives, i.e., TAA, TABA, TADA, and TAH—as well as the samples with different ratios—clearly showed the formation of H-bonding crosslinks in the solvent-free network. This was confirmed by the band wavenumbers of amide groups ($\upsilon_{C=O}$ ~ 1650 cm$^{-1}$; $\upsilon_{N-H}$ ~ 3300 cm$^{-1}$)[36]. In the case of poly(TAA-$S_x$) copolymer, it was found that the amide band shifted via the different ratios of the two monomers (Supplementary Fig. 30), suggesting the possible formation of S-mediated H-bonds in the sulfur-rich copolymer network. These weak yet abundant H-bonds could facilitate the mechanical stretchability and toughness of the supramolecular crosslinked network by acting as sacrificial non-covalent bonds. Rheological measurements further confirmed the typical heat-induced flowing behavior of a supramolecular crosslinked network (Supplementary Figs. 35 and 36). The resulting supramolecular crosslinked network exhibited a glass-transition temperature of 23 °C and a flow temperature of 90 °C, suggesting that the product is a thermoplastic material.

Considering the dynamic nature of the ROP, we further investigate how the minor residues of unpolymerized monomers (around 10% according to $^1$H NMR analysis; Fig. 2c) make an effect on the material properties. The poly($S_8$/TAA) copolymer was purified by immersing in $CS_2$ for three days to remove soluble species (e.g., monomers, oligomers, etc.) We obtained a pinky-white polymer solid after drying in a vacuum, which was characterized by TGA, DSC, solid-state UV–Vis spectroscopy, FT-IR, and NMR (Supplementary Fig. 37). Solid-state UV–Vis spectra showed a decrease in the absorption peak of the oligomeric $S_x$ peaks at 450 and 520 nm after washing with $CS_2$. TGA indicated similar thermal decomposition temperatures before and after purification. DSC showed a slightly decreased $T_g$ after purification, suggesting that the monomer residues are not plasticizers for the bulk copolymers. These results suggested that, even if this dynamic ROP reaction does not produce pure copolymers after bulk copolymerization, the material preparation is still free of purification as most dynamic polymeric materials are based on poly(disulfide)s.

## Mechanical properties

We then investigated the mechanical performance of the resulting copolymers by uniaxial tension tests. The stress–strain curves show that the mechanical properties of the copolymers could be finely tuned over a large range, including soft materials (0.9–9.1 MPa), elastomers (68.1–214.9 MPa), and rigid plastics (224.9–949.1 MPa) in Young's moduli (Fig. 3a, b and Supplementary Figs. 38–42) simply based on copolymerization ratios and sidechain groups. The effect of the copolymerization ratio was investigated by comparing the mechanical tensile curves of the copolymers with different monomer ratios (Supplementary Fig. 38). It was found that the copolymer network was stiffened and strengthened with the increase of sulfur content. For example, The Young's modulus of poly($S_8$/TAA) was increased from 68 MPa ($S_8$:TAA = 1:5) to 215 MPa ($S_8$:TAA = 1:1), while the Young's modulus of poly($S_8$/TABA) was increased from 4.2 MPa ($S_8$:TABA = 1:3) to 9.1 MPa ($S_8$:TABA = 1:1). This feature of mainchain-structure-dependent mechanical properties enables tunability of the resulting copolymers by controlling the copolymerization ratios of the two monomers, i.e., cyclic disulfides and elementary sulfur.

On the other hand, the sidechain of cyclic disulfides also provides versatile tunability by engineering the supramolecular crosslinking interactions. For example, poly(TAA-$S_x$) showed typical elastomer-like stress–strain curve with ductility (135.0–1153.1%) and good mechanical toughness (6.5–14.6 MPa) due to the existence of abundant H-bonds of primary amide groups acting as sacrificial bonds (Fig. 3c and Supplementary Fig. 39). Poly(TAH-$S_x$) exhibited a high Young's modulus of 224.9–949.1 MPa as a result of the highest density of sidechain H-bonds (Fig. 3c and Supplementary Fig. 40). The maximum stress strength was 21.1 MPa, which was four-fold the strongest poly(TAA-$S_x$). For reference, poly(TABA-$S_x$) and poly(TADA-$S_x$) were gel-like soft materials with very dynamic and creeping properties at room temperature (Supplementary Figs. 41 and 42), which might be interesting for wearable devices and flexible sensors. The activation energy ($E_a$) of poly(TAA-$S_x$) and poly(TAH-$S_x$) is higher than that of poly(TABA-$S_x$) and poly(TADA-$S_x$), which further confirmed that the supramolecular crosslinks work on the solvent-free H-bonding network (Supplementary Figs. 43–46). Therefore, sidechain modification offers great chemical spaces and opportunities to engineer supramolecular crosslinks as well as possible functions for applications.

The existence of abundant H-bonds and disulfide bonds in the solvent-free network gives the copolymer self-healing properties[34]. A piece of poly(TAA-$S_x$) film was cut into two fragments. These were then interfaced at room temperature, thus yielding a repaired sample in 12 h with recovered mechanical tensile properties versus the original sample (Fig. 3d). We attribute the high self-healing efficiency to the dynamic crosslinking of supramolecular interactions, i.e., H−bonds. Despite the dynamic covalent nature of the disulfide bonds, it seems that H-bonds are mainly responsible for the interface repair process because the poly($S_x$-DIB) copolymers exhibit no self-healing ability[17]. Meanwhile, poly(TAH-$S_x$) copolymers also showed good processability and allowed the preparation of thin fibers from molten liquid (Fig. 3e).

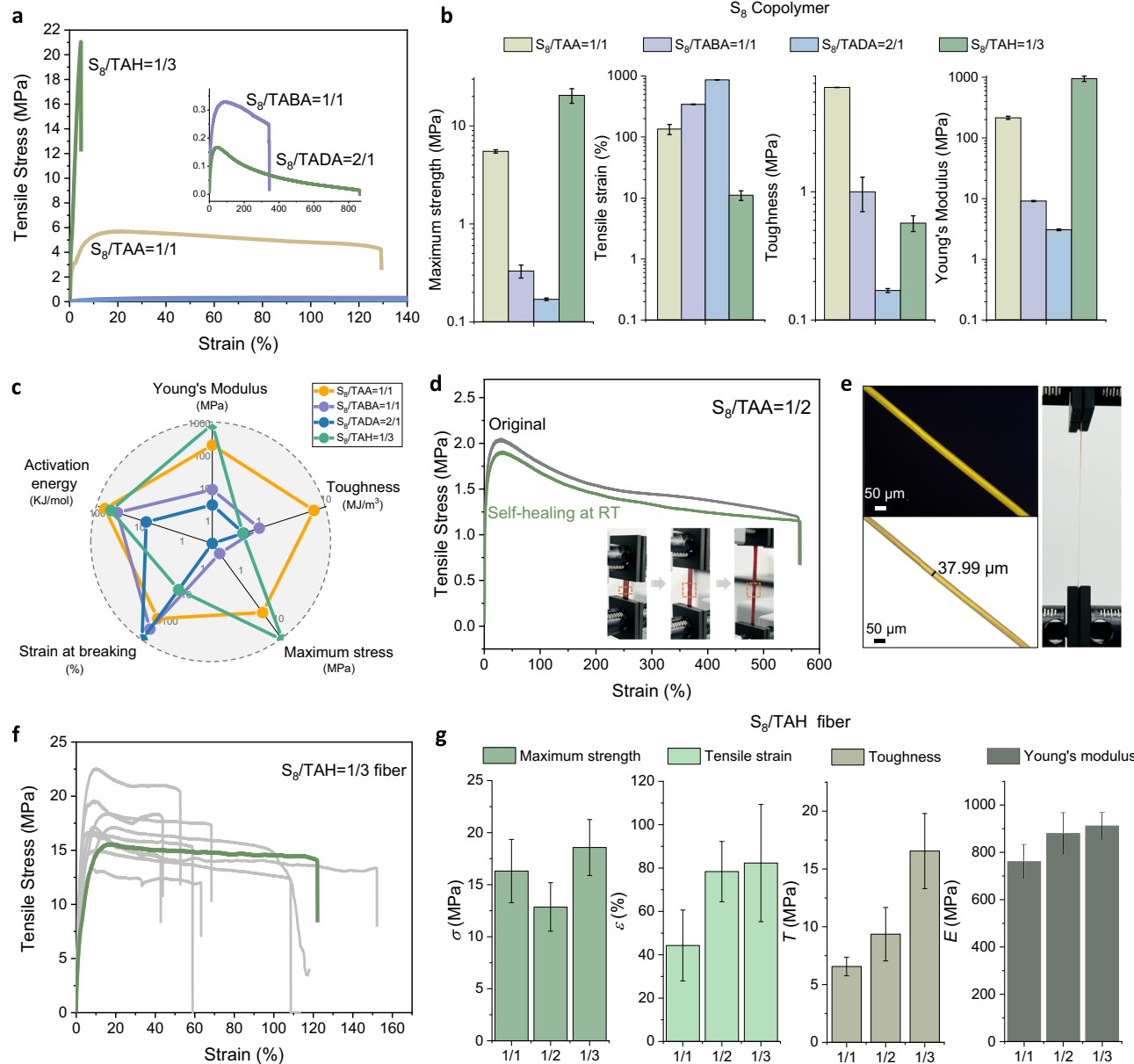

**Fig. 3 | Mechanical properties of sulfur-rich copolymers. a** Stress–strain curves of the resulting $S_8$ copolymers with various disulfide monomers. **b** A summary of mechanical performances of the $S_8$ copolymers. Measurements were carried out in triplicate ($n = 3$); error bars show the standard deviation of the replicate measurements, the error bar centers are the means of the replicate measurements. **c** Radar map showing the effect of side chain functional group on the mechanical performance. **d** Stress–strain curves of poly($S_8$/TAA = 1/2) before and after self-healing at room temperature after 12 h. RT room temperature. **e** Optical microscopy image (left) and photograph during tensile test of poly($S_8$/TAH = 1/3) filament. The scale bar is 50 µm. **f** Stress–strain curves of poly($S_8$/TAH = 1/3) filament. **g** A summary of mechanical performances of poly($S_8$/TAH) filament with different amounts of TAH monomer. Measurements were repeated with at least five independent samples, error bars show the standard deviation of the replicate measurements, the error bar centers are the means of the replicate measurements. $\sigma$: maximum strength, $\varepsilon$: temsile strain, $T$: toughness, $E$: Young's modulus.

The resulting fibers were mechanically stretchable and tough, as shown by stress–strain curves with a high Young's modulus (more than 0.8 GPa), good stretchability (over 100%), high strength (15 MPa engineering stress), and high mechanical toughness (9.4 MPa) (Fig. 3f, g, Supplementary Fig. 47, and Table 1). Further comparison with the previously developed poly(sulfur) materials demonstrated the state-of-the-art mechanical performance within the poly(sulfur)-based rubbery elastomers (Supplementary Table 2 and Fig. 48).

## Adhesion performance

Despite the prior developments of poly(sulfur)-based elastomers, their use as adhesives has not yet been explored to the best of our knowledge. Our copolymer materials are based on two types of small-molecule feedstock and could be readily cured in situ on substrates, sticking to three surfaces with high adhesion strength. The key feature of this small-molecule adhesive rests on its ability to penetrate the rough and microscopically porous surface to reach a maximum area for interfacial adhesion. Meanwhile, the sidechain sticky groups, i.e., amide and acylhydrazine groups, can also offer anchoring sites for diverse surfaces (Fig. 4a). Three substrates commonly used in engineering applications, e.g., stainless steel, glass, and aluminum, were used to evaluate our copolymer adhesives. Samples with different anchoring groups were compared (Fig. 4b). Not unexpectedly, poly($S_x$-TAA) and poly($S_x$-TAH) copolymers with sticky H-bond anchoring

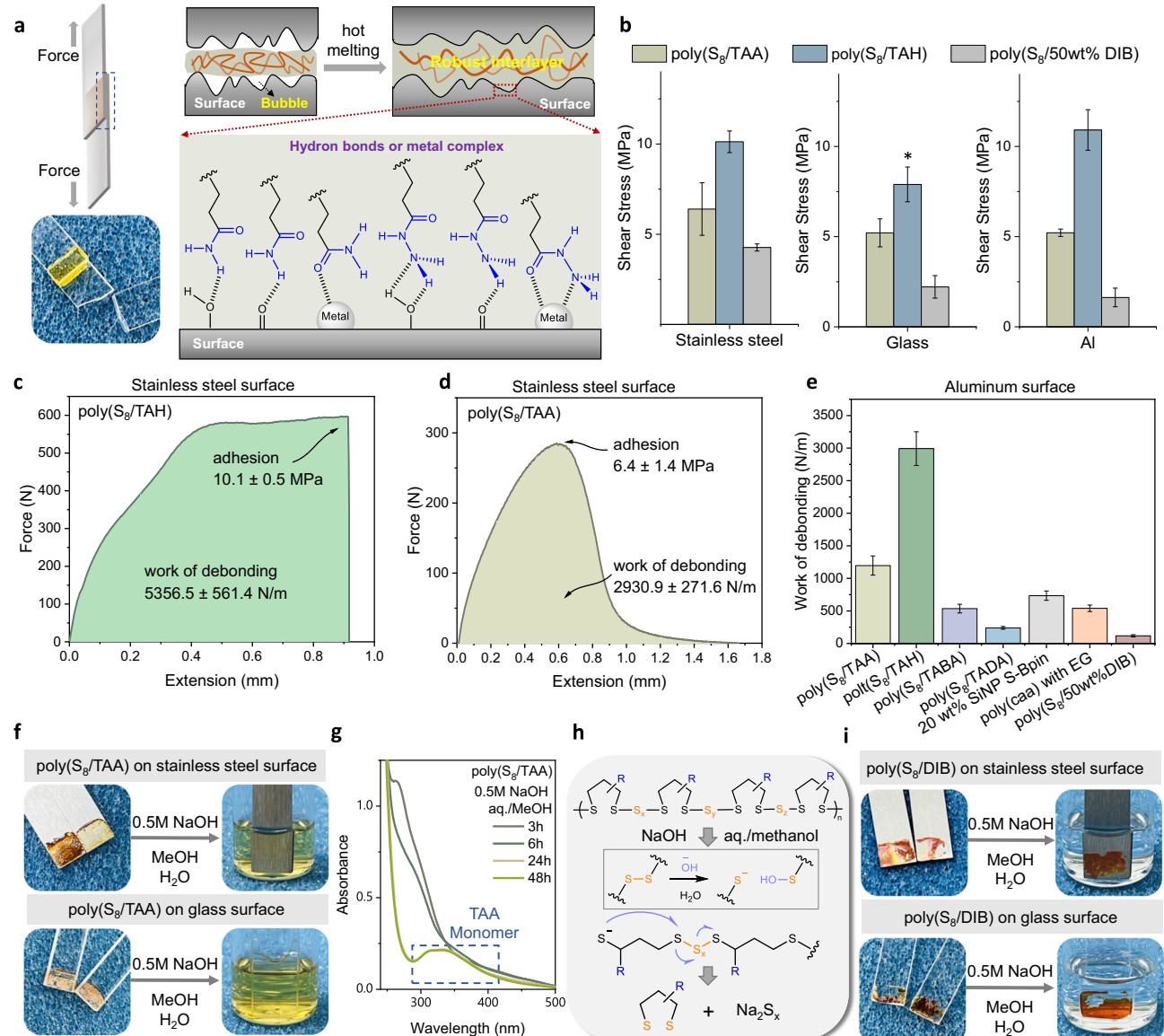

**Fig. 4 | Interfacial adhesion strength of S₈ copolymers. a** Schematic representation of the adhesion behavior. **b** The shear strength of S₈ copolymers with various disulfide monomers for a variety of surfaces (* means that the use of poly(S₈/TAH) caused the failure of the glass substrates before breaking of the adhesive bond; see Supplementary Movie 1). Measurements were carried out in triplicate ($n = 3$); error bars show the standard deviation of the replicate measurements, the error bar centers are the means of the replicate measurements. Al: aluminum. Force-versus-extension curve for **c** poly(S₈/TAH = 1/2) and **d** poly(S₈/TAH = 1/2) shows work of debonding and lap shear adhesion strength on stainless steel. **e** Debonding of poly(S₈/TAA), poly(S₈/TAH), poly(S₈/TABA), poly(S₈/TADA), SiNP-Bpin[37], poly(caa) with EG[38], and poly(S₈/50%DIB) on aluminum surface.

Measurements were carried out in triplicate ($n = 3$) error bars show the standard deviation of the replicate measurements, and the error bar centers are the means of the replicate measurements. Results from the literature were extracted from corresponding papers. **f** Adhesives of poly(S₈/TAA) on stainless steel and glass surfaces are degradable in base solution within hours at room temperature. **g** The detected UV–Vis spectra of the degraded poly(S₈/TAA) catalyzed by base. **h** Prosed degradation mechanism of the polymer network in an aqueous/methanol mixture of alkaline. **i** The poly(S₈/50 wt% DIB) shows poor depolymerizability in the base solution even after 90 days. The base solution is an aqueous/methanol mixture with NaOH at a concentration of 0.5 M.

groups exhibited the strongest adhesion performance with shear strength over 10 MPa.

More remarkable results are observed for surface release, i.e., debonding. These data were calculated from the stress–strain curves of the adhered samples[37,38]. The resulting adhesive interlayers of poly(Sₓ-TAA) and poly(Sₓ-TAH) were not only strong but also extremely tough (Fig. 4c and d). Poly(Sₓ-TABA) and poly(Sₓ-TADA) are much weaker and ductile (Fig. 4d and Supplementary Fig. 49). For example, poly(Sₓ-TAA) debonds at 2.93 kN/m with a high shearing strength of 6.4 MPa. The poly(Sₓ-TAH) debonds up to 5.36 kN/m on steel surfaces, which is the highest value yet reported (Fig. 4e). The H-bond

crosslinkers act as sacrificial bonds for energy dissipation. This high resistance towards de-bonding might also be attributed to the high-density sulfur–metal interactions because the control sample poly(Sₓ-DIB) also bears moderate adhesion strength (Supplementary Figs. 50, 51 and Table 3).

Moreover, the polymeric backbone rich in sulfur–sulfur bonds enables the resulting adhesive materials to be degradable and removable[39,40]. The adhesive layers show solubility in the polar organic solvent, e.g., DMSO, DMF, hexamethylphosphoramide (Supplementary Figs. 52–54), while immersing the adhesive layers into the base solution at room temperature induces the spontaneous

depolymerization of the network into monomers and oligomers in solutions (Fig. 4f). The degradation process can be monitored by UV−vis spectroscopy, thus showing the recovery of the distinctive absorption band of cyclic disulfides (Fig. 4g, h and Supplementary Fig. 55). As a control, the poly($S_x$-DIB) copolymers were not degradable under the same condition (Fig. 4i and Supplementary Fig. 55) mainly due to their covalently crosslinked nature. Thus, the thermoplastic copolymers presented here can be used as high-performance hot-melt adhesives that are degradable and sustainable.

## Discussion

In summary, the combination of two small molecules, elementary sulfur, and thioctic acid derivatives, could generate thermoplastic copolymers with tunable mechanical properties. The resulting materials exhibit typical rubber-like elastomeric properties as well as self-repairing ability at room temperature. The synergy of sidechain groups and sulfur-rich main chains gives the copolymers extremely high adhesion toughness. The dynamic covalent backbone also enables degradability under mild conditions. We envision that the discovery of this material could promote the sustainable development of sulfur-based polymers−especially toward the applications in self-healing elastomers, high-performance adhesives, and degradable plastics.

## Methods

### Preparation of TA-based derivatives monomers

A general preparation method is as follows: (±)-α-thioctic acid (TA) (1.0 eq, 4.8 g, 23.0 mmol) was dissolved in $CH_3CN$ (200 mL). Di-(N,N′-succinimidyl) carbonate (1.2 eq, 7.2 g, 28.0 mmol) and triethylamine ($Et_3N$, 3.0 eq, 10.0 mL) were added into the above solution and the mixture was stirred at room temperature for 2 h. The mixture was concentrated at 35 °C under reduced pressure and then added to an aqueous 5% $NaHCO_3$ solution to produce yellow precipitates (TA-NHS). After dissolving of TA-NHS in $CH_3CN$ (200 mL), ammonia solution, hydrazine monohydroxide, n-butylamine, or diethylamine was added into the mixture to form side-chain modified TA-based derivatives. The details of after treatment are in the Supporting Information (Experimental method).

### Polymerization procedure

In a typical procedure, 2 g (7.8 mmol) of $S_8$ monomer was added into a Teflon vial, which was then heated by a metal heating block at a constant temperature of 120 °C. Yellow viscous liquid $S_8$ was obtained under magnetic stirring. In the case of polymerization, a given amount of co-monomer was added to the molten $S_8$ liquid and dissolved by vigorous stirring. Next, the reaction mixture was stirred at 120 °C for 2 h to obtain homogeneity, and the resulting liquid was quickly transferred into a Teflon mold and cooled to room temperature to form free-standing polymer samples.

### Polymer characterization

Powder X-Ray diffraction (PXRD) experiments were undertaken on a Rigaku rotating anode X-ray powder diffractometer (18 kW/D/max2550VB/PC) equipped with a copper target 18 kW (450 mA), a fully automated curved (plate) crystal graphite monochromator and a programmed variable slit system. The morphology and element distribution of the polymer were determined on the field emission scanning electron microscopy (FESEM, GeminiSEM 500). Attenuated total reflection spectroscopy (ATR) data was performed on a Thermo Fisher instruments Nicolet iS50R FT-IR Spectrometer. Solid polymer samples were added onto the surface of the sample platform. The background of the sample platform was corrected. The measurements were performed at room temperature (25 °C). Differential scanning calorimetry (DSC) data were carried out on a TA instrument DSC Q1000 in a dry nitrogen atmosphere. Samples were cycled from −50 to 100 °C at a rate of 10 °C/min. Glass transition temperature was found by taking the

midpoint of the reversible endotherm of the second heating scan for each sample. Thermogravimetric analysis (TGA) data were carried out on a TGA5500 (TA Instruments, USA) in a nitrogen atmosphere with a temperature range from 20 to 600 °C with a heating rate of 10 °C/min. Matrix-assisted laser desorption/ionization time-of-flight mass spectrometry (MALDI-TOF-MS) mass spectra were recorded on an ABS 4800 plus MALDI-TOF/TOF (400−500,000 Da). X-ray photoelectron spectroscopy (XPS) was performed using a Thermo Fisher ESCALAB 250Xi with a monochromatic Al Kα X-ray source. Raman Analysis (Raman) was carried out on a Laser Micro-Raman Spectrometer (≤10.2/cm/invia reflex) with high-performance grade Leica DMLM microscope at room temperature.

Rheology measurements were performed on a TA Instruments Discovery HR-2 rheometer. Polymer samples were placed under a 20-mm-diameter parallel plate with a gap of 0.1–2.0 mm. Temperature dispersion tests were carried out in a range from −5 to 150 °C (5 °C/min) at a constant frequency of 1 Hz with an applied strain of 0.09%. Frequency sweep tests were conducted in a range from 626 to $6.28 \times 10^3$ rad/s at room temperature with an applied strain of 0.09%. Creep recovery tests were carried out between 10 and 20 °C by imposing a constant stress over time (creep) and releasing it for a certain period of time (recovery) while measuring the strain. Stress relaxation analysis tests for poly($S_8$/TAA = 1/2) were performed between 25 and 45 °C with a constant strain of 0.09%. Stress relaxation analysis tests for poly($S_8$/TABA = 1/2) were performed between 25 and 45 °C with a constant strain of 0.2%. Stress relaxation analysis tests for poly($S_8$/TADA = 1/2) were performed between 5 and 25 °C with a constant strain of 1.5%. Stress relaxation analysis tests for poly($S_8$/TAH = 1/2) were performed between 25 and 45 °C with a constant strain of 0.05%. The relaxation times were from a plot of the normalized relaxation modulus reaching $1/e$ (37%). The apparent activation energy ($E_a$) was measured from the fitted curves of $\ln(\tau)$ vs. $1000/T$ according to the Arrhenius equation:

$$\ln\tau = \ln\tau_0 + E_a/RT \qquad (1)$$

where $\tau$ is the relaxation time when the value of $G_t/G_0$ reaches $1/e$, $R$ is the gas constant (8.314 J K$^{-1}$ mol), and $T$ is the thermodynamic temperature.

### Preparation of adhesive

Copolymers were directly applied as dry adhesives. The polymers were molten into low-viscosity liquid by heating at 120 °C and then deposited on the surfaces to adhere two surfaces together after cooling down to room temperature.

### Tensile analysis

Tensile tests of copolymers were measured using an Instron 34TM-5 universal testing system equipped with a 100 N sensor. The data were recorded in real time by a wire-connected computer system. Copolymer films were cut into tensile bars with a width of 5 mm, a length of 50 mm, and an average thickness of 1 mm. Specimens were tested at a fixed tensile speed of 50 mm/min. Tensile measurement was carried out at ambient conditions for all samples, and each measurement was repeated with at least three independent specimens. For the tensile tests of fibers, the tensile speed was fixed at 10 mm/min and each measurement was repeated with at least 10 independent specimens.

### Lap shear testing

Aluminum and stainless-steel substrates were commercially available rectangular pieces with a length of 50 mm, a width of 10 mm, and a thickness of 2 mm. The toughened glass was commercially available in rectangular pieces with a length of 50 mm, a width of 10 mm, and a thickness of 3 mm. Adhesion tests were performed at room temperature on an Instron 34TM-5 tensile machine mounted with a 5 kN load

cell. The adhered samples were fixed to the jigs of the tensile machine to measure the lap shear strength of the adhesion layer. In the case of glass surfaces, special ultra-strong glass plates were used to be able to sustain the shearing force due to the high adhesion strength of the adhesive interfaces.

Lap shear adhesion is defined as the maximum force (in N) of the adhesive joint divided by the overlap area (in mm$^2$).

$$\text{Lap shear strength} = (\text{Force}(N))/(\text{Adhesive area}(mm^2)) \qquad (2)$$

Work of debonding is defined as the integrated area under the force-versus-extension curve.

$$\text{work of debonding} = \int_0^x F dx / A_{\text{bonded overlap}} \qquad (3)$$

where $\int_0^x F dx$ is defined as the integral of the force vs. extension curve, or energy of adhesion (in J) and $A_{\text{bonded overlap}}$ is the measured overlap area of the adhesive joint (in m$^2$).

## Data availability

The authors declare that the data supporting the findings of this study are available within the paper and its Supplementary Information files or from the corresponding author upon request.

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

## Acknowledgements

This work was supported by the National Natural Science Foundation of China (grant nos. 22220102004, 22025503), Shanghai Municipal Science and Technology Major Project (grant no. 2018SHZDZX03), the Fundamental Research Funds for the Central Universities, the Program of Introducing Talents of Discipline to Universities (grant no. B16017), Science and Technology Commission of Shanghai Municipality (grant no. 21JC1401700), the Starry Night Science Fund of Zhejiang University Shanghai Institute for Advanced Study (grant no. SN-ZJU-SIAS-006), Shanghai Pujiang Program (grant no. 23PJ1402100), China Postdoctoral Science Foundation (grant no. 2022TQ0105), the National Natural Science Foundation of China (grant no. 22305083), Shanghai Pujiang Program (grant no. 23PJD020), the European Research Council (ERC; advanced grant no. 694345 to B.L.F.), the Dutch Ministry of Education, Culture and Science (Gravitation program No.024.601.035).

## Author contributions

Y.D. and Q.Z. conceived the project. Q.Z., D.-H.Q. and B.L.F. supervised the research. Y.D. and Z.H. carried out the synthesis, characterizations, and data acquisition. Y.D., Q.Z., B.L.F., D.-H.Q. and H.T. analyzed the data and wrote the manuscript.

## Competing interests

The authors declare the following competing interests: Y.X.D., Z.H., Q.Z., and D.-H.Q. are inventors on a provisional patent application related to this work that has been filed by the East China University of Science and Technology (serial no. 2023113880742, date: 25 October 2023). The authors that are not named in the patent declare no other competing interests.
