## [Peer Review File · Nature Communications]

REVIEWER COMMENTS

Reviewer #1 (Remarks to the Author):

The manuscript by Deng et al. reports on the copolymerization of lipoic acid derivatives with elemental sulfur to prepare organopolysulfides along with demonstration of these materials for certain uses and adhesives. The chemical concept and demonstration of copolymerization of lipoic acid based comonomers and S₈ is novel and the properties of the resulting materials look reasonable. Hence, the work and manuscript have significant merit and interest to the readers of Nature Commun. However, the structural characterization of polymers is lacking which points to potential issues of partial unwanted sulfuration of carbonyl moieties, along with incomplete thermal characterization of these materials. Hence, it is recommended that the manuscript undergo the following major revisions to address the concerns below:

- 1) due to the broad scope of lipoic acid esters and amides prepared for copolymerization in liquid sulfur, it would be important to show that all of these monomers can be made miscible in liquid sulfur at the appropriate temperatures for polymerization. The COOH and hydrazide based monomers in liquid sulfur would be the most important monomers so show this is the case.
- 2) the carbonyl moieties in the lipoic acid based esters and amides are highly susceptible to sulfuration, or partial sulfuration which would be easy to miss in IR spectroscopic characterization. The authors should provide for these sulfur polymers (if soluble) ¹³C NMR spectroscopy comparing the monomer vs the sulfur copolymer to show the C-carbonyl atoms are preserved, or (partially) sulfurated. If the polymers are insoluble, then the use of miscible model compounds would be sufficient evidence.
- 3) the polymer architecture of these sulfur copolymers needs to be clarified, or proposed. S₈ with a mono-lipoic acid monomer should afford linear polymers, unless CH₂ H-abstraction is present as recently shown is a major side reaction in liquid sulfur (JACS 2023,145, 12386).
- 3) The thermal properties (T_g, T_m, T_c) for these sulfur copolymers should be clearly tabulated somewhere (SI would be likely) and mentioned in the text. It is completely unclear at this point what trends in T_g result in monomer side chain variation.

On this note, the authors suggest there are supramolecular non-covalent associations between sulfur copolymers containing amide groups, which is where DSC T_g would be important to cross-check with the IR spectroscopic data of N-H hydrogen bond shifting.

- 4) are any of the sulfur copolymers soluble? What are the observations and trends with polar-COOH and other groups? ¹H NMR is provided, so presumably this can be explained

In summary, this referee is interested and excited about this report. BUT-given the large amount of poorly reported structural characterization in this field, it is paramount that new methods for sulfur polymerization adequately include sufficient chemical characterization of sulfur copolymer microstructure, polymer architectures, basic bulk properties (solubility, T_g/T_c/T_m).

Reviewer #2 (Remarks to the Author):

Elemental sulfur is one of the most abundant elements in the Earth. Meanwhile, millions of tons of sulfur is produced as a by-product from petrochemical industry each year. Elemental sulfur utilization has been a global issue. Using sulfur (S₈) as a monomer to prepare polymer materials is a good strategy but still faces challenges. The manuscript by Deng et al reports copolymerization of S₈ with cyclic disulfides, which converts elementary sulfur into sulfur-rich thermoplastics. The polymer exhibits elasticity and self-repairing ability at room temperature. Because of the synergy of sidechains and sulfur-rich main chains, it has high adhesion toughness. The polymer also has degradability under mild conditions due to the dynamic covalent backbone. This report provides a simple but efficient approach to develop functional sulfur containing polymers. It makes significance for utilization of sulfur. Generally, the manuscript could be published in Nature Communications after a revision. However, the following points should be taken into consideration.

1)The ring-opening polymerization of cyclic disulfides can yield high-molecular-weight sulfur-containing polymer as reported before (J. Am. Chem. Soc. 2019, 141, 43, 17075; 2013, 135, 6, 2088; 2014,136, 6069). However, the copolymerization of S₈ with cyclic disulfides only results oligomer reported here. Why? More detailed the polymerization mechanism should be given.

2)The residual of monomers has profound effect on the mechanical performance, so the product should be purified, and the monomer conversion should be given.

3)The two-stage degradation in TGA curves (Figure S25) should be explained.

4)TAME should be explained in the main text.

5)The description in Line 51-52 is not correct. Anionic hybrid copolymerization is quite different from the anionic ring-opening polymerization in mechanism, the former is absolutely not the expansion of the latter. The sentence is suggested to correct as follows: The anionic ring-opening polymerization of S₈ was first reported by Penczek et al. in 1978. Very recently, Yang et al. discovered the anionic hybrid copolymerization of S₈ with acrylate.

Reviewer #3 (Remarks to the Author):

Qu, Zhang and coworkers have presented a clever use of strained dithiolanes to recruit otherwise wasted sulfur to manufacture new materials. The synthesis is green, free of catalyst or solvent. The wide scope of monomer lends the materials a wide range of accessible property parameters. As a demonstration, the polymer is shown to be elastic, self-healable, adhesive, and degradable. I strongly support the publication of this manuscript. I have some technical questions that I hope the authors can offer to clarify:

1) Page 3 Line 62 / Figure 1. The authors proposed a radical pathway. Is it possible that S-S can undergo direct exchange with S8? In other words, if radical quencher is added / present, will the polymerization still be robust? I ask this question, because I imagine the reaction would have a broader impact or utility if the reaction is tolerant with impurities.

2) Page 3 Line 65, I guess the use of “network” is not justified? And the following sentence seemed a bit confusing. Did the author intend to say the reactions is promoted by ROP?

3) Figure 2c. There are some unidentified peaks in the polymer sample, e.g., ~1.8 ppm. The reaction between S8 and TAA is supposed to be clean and the starting materials looks clean, too. Then, can the author comment on what are the other species present in the polymer sample? This is important, since the authors claimed the free of need to purify.

4) Figure 2d, top panel. The boundary between the green and blue regions seem arbitrary? I wonder if some reference spectrum or its assignment should be provided in the SI. Also, I don't feel like broadening is the right word for C-S vibration, it simply nearly disappeared in the polymer. Stating that it is broadened appears to be a cyclic argument to me: that you assume it is the compound, and therefore it should be broadened?

5) Page 7 Line 116: These are different in terms of bond connectivity – feels like a strange statement to me. I assume the authors meant the spectra is different because of changes in connectivity?

6) Page 7 Line 119 / Fig S22-S24: The MALDI showed different patterns in S_x segment. Does that mean different monomers led to polymers of various multi-S segments? Then how does the XPS corroborate with the length of S in the polymer, and how about different S_x / monomer ratio? How about mechanical and other properties shown in this manuscript? Is a long S_x better or worse, for example. I felt this is probably the most important issue that is kind of weak in this paper. I hope the material properties can be better correlated with the structures.

7) Page 9 Line 165: “Systematic investigations of the copolymer samples with different ratios revealed mechanical tunability of the materials” – I think the authors can be more quantitative here since they have the numbers, include the sentence that follows. What is the range for the state-of-the-arts?

8) Page 13, Line 212 / Figure 4h: What is the role of DMSO and how does it degrade the polymer? What about other polar solvent such as DMF and HMPA?

Point-by-point Response to Reviewers' Comments

Reviewer #1 (Remarks to the Author):

The manuscript by Deng et al. reports on the copolymerization of lipoic acid derivatives with elemental to prepare organopolysulfides along demonstration of these materials for certain uses and adhesives. The chemical concept and demonstration of copolymerization of lipoic acid based comonomers and S₈ is novel and the properties of the resulting materials look reasonable. Hence, the work and manuscript have significant merit and interest to the readers of Nature Commun. However, the structural characterization of polymers is lacking which points to potential issues of partial unwanted sulfuration of carbonyl moieties, along with incomplete thermal characterization of these materials. Hence, it is recommended that the manuscript under the following major revisions to address the concerns below:

We appreciate the positive comments and suggestions of this referee. Additional experiments have been performed to address the issues mentioned, and new data have been added to the revised manuscript and SI.

1) due to the broad scope of lipid acid esters and amides prepared for copolymerization in liquid sulfur, it would be important to show these all of these monomers can be made miscible in liquid sulfur at the appropriate temperatures for polymerization. The COOH and hydrazide based monomers in liquid sulfur would be the most important monomers so show this is the case.

We have added a set of new photograph pictures (Fig. S8) to show the miscibility of the mentioned monomers in liquid sulfur at 120°C.

Supplementary Fig. 8 | Copolymerization procedure of S₈ and TA-based derivatives at 120°C. The weight ratio of S₈ to monomer is 1 to 1.

2) the carbonyl moieties in the lipoic acid based esters and amides are highly susceptible to sulfuration, or partial sulfuration which would be easy to miss in IR spectroscopic characterization. The authors should provide for these sulfur polymers (if soluble) ^{13}C NMR spectroscopy comparing the monomer vs the sulfur copolymer to show the C-carbonyl atoms are preserved, or (partially) sulfurated. If the polymers are insoluble, then the use of miscible model compounds would be sufficient evidence.

We exclude the possible sulfuration reaction by performing additional ^{13}C NMR spectroscopic measurements of poly(S_8/TAMe) and poly(S_8/TAA) dissolved in $\text{DMSO-}d_6$, showing i) the consistency of the carbonyl carbon peak (C_h at 173 ppm) between monomers and polymers, and ii) the absence of sulfurated carbon peak. The discussions have been added to the revised manuscript.

Supplementary Fig. 18 | ^{13}C NMR spectrum of compound TAMe and poly(S_8/TAMe) ($\text{DMSO-}d_6$, 151 MHz, 298 K).

Supplementary Fig. 19 | ^{13}C NMR spectrum of compound TAA and poly(S_8/TAA) ($\text{DMSO-}d_6$, 151 MHz, 298 K).

3) the polymer architecture of these sulfur copolymers needs to be clarified, or proposed. S_8 with a mono-lipoic acid monomer should afford linear polymers, unless CH_2 H-abstraction is present as recently shown is a major side reaction in liquid sulfur (JACS 2023,145, 12386).

The polymer architecture of these sulfur copolymers has been proposed as linear polymers accordingly in the revised manuscript.

4) The thermal properties (T_g , T_m , T_c) of these sulfur copolymers should be clearly tabulated somewhere (SI would be likely) and mentioned in the text. It is completely unclear at this point what trends in T_g result in monomer side chain variation. On this note, the authors suggest there are supramolecular non-covalent associations between sulfur copolymers containing amide groups, which is where DSC T_g would be important to cross-check with the IR spectroscopic data of N-H hydrogen bond shifting.

Additional DSC characterizations of copolymers have been added to the revised manuscript. It is found that, as expected, the T_g values show a relationship with sidechain interactions reflected by IR data, i.e., stronger H-bonds give higher T_g .

Fig. 2 | Structure characterization of poly(S_8 /TAA) copolymers. f, DSC data of copolymers.

5) are any of the sulfur copolymers soluble? What are the observations and trends with polar-COOH and other groups? 1H NMR is provided, so presumably this can be explained. We have included the solubility test in Fig. S21. Not surprisingly, polar sidechain groups rich in H-bonds give lower solubility of the resulting copolymers. All the samples are dissolvable in DMSO and DMF. We have added additional description about the solubility into the revised manuscript for clarification.

Supplementary Fig. 23 | Solubility tests of different copolymer samples. Solubility test of the resulting copolymers. (A) Photographs of the polymer samples soaked in different solvents for 24 h; (B) Tables of the relationships between solvent solubility and copolymers. (“√” means totally soluble; “Δ” means swelling; “x” means insoluble).

In summary, this referee is interested and excited about this report. BUT-given the large amount of poorly reported structural characterization in this field, it is paramount that new methods for sulfur polymerization adequately include sufficient chemical characterization of sulfur copolymer microstructure, polymer architectures, basic bulk properties (solubility, $T_g/T_m/T_c$).

We very much appreciate the positive comments and the constructive suggestions of this reviewer. We hope the current version of the manuscript, with the added characterization data as requested supported our findings, is in a good shape of publication.

Reviewer #2 (Remarks to the Author):

Elemental sulfur is one of the most abundant elements in the Earth. Meanwhile, millions of tons of sulfur is produced as a by-product from petrochemical industry each year. Elemental sulfur utilization has been a global issue. Using sulfur (S_8) as a monomer to prepare polymer materials is a good strategy but still faces challenges. The manuscript by Deng et al reports copolymerization of S_8 with cyclic disulfides, which converts elementary sulfur into sulfur-rich thermoplastics. The polymer exhibits elasticity and self-repairing ability at room temperature. Because of the synergy of sidechains and sulfur-rich main chains, it has high adhesion toughness. The polymer also has degradability under mild conditions due to the dynamic covalent backbone. This report provides a simple but efficient approach to develop functional sulfur containing polymers. It makes significance for utilization of sulfur. Generally, the manuscript could be published in Nature Communications after a revision. However, the following points should be taken into consideration.

We appreciate the positive comments and valuable suggestions of this referee. Additional experiments have been performed to address the issues mentioned, and new data have been added to the revised manuscript and SI.

1)The ring-opening polymerization of cyclic disulfides can yield high-molecular-weight sulfur-containing polymer as reported before (J. Am. Chem. Soc. 2019, 141, 43, 17075; 2013, 135, 6, 2088; 2014,136, 6069). However, the copolymerization of S_8 with cyclic disulfides only results oligomer reported here. Why? More detailed the polymerization mechanism should be given.

It is noted that we didn't claim the oligomeric nature of our copolymers. We use the term "oligomeric" to refer to the S_x part. In this manuscript, we don't aim at comparing with the homo-polymerization of 1,2-dithiolanes due to the different chemical nature. Regarding the polymerization mechanism, our system is proposed as a thermal radical pathway.

2)The residual of monomers has profound effect on the mechanical performance, so the product should be purified, and the monomer conversion should be given.

This is a valuable suggestion and we have attempted to purify our copolymer by immersing it in CS_2 for three days to remove soluble species (e.g., monomers, oligomers, etc.) We obtained a pinky-white polymer solid after drying in a vacuum, which was characterized by TGA, DSC, solid-state UV-Vis spectroscopy, FT-IR, and NMR (Supplementary Fig. 37). Solid-state UV-Vis spectra showed a decrease in the absorption peak of the oligomeric S_x peaks at 450 nm and 520 nm after washing with CS_2 . TGA indicated similar thermal decomposition temperature after purification. DSC showed a slightly decreased T_g after purification, suggesting that the monomer residues are not plasticizers in the bulk copolymer network. These results suggested that, even if this dynamic ROP reaction doesn't produce pure copolymers after bulk copolymerization, the material preparation is still free of purification as most dynamic polymeric materials based on poly(disulfide)s. The

monomer conversion is estimated as 90% according to ^1H NMR analysis. All these new data and discussions have been added to the revised manuscript and SI.

Supplementary Fig. 37 | Purification with CS_2 . (A) The images of the purification process. (B) ^1H NMR spectrum, (C) Solid-state UV-Vis spectra, (D) TGA analysis, (E) DSC analysis, and (F) ATR analysis of the $\text{poly}(\text{S}_8/\text{TAA}=1/2)$ after purification.

3)The two-stage degradation in TGA curves (Figure S25) should be explained.

The first stage is due to the thermal decomposition of organic cyclic disulfides. The observation of the second stage is attributed to the inorganic sulfur in the product. The description has been added to the supplementary information.

4)TAMe should be explained in the main text.

The resulting copolymer of $\text{poly}(\text{S}_8/\text{TAMe})$ shows a viscous and flowing liquid behavior because of the lack of supramolecular sidechain crosslinking (Supplementary Fig.9). The description has been added to the revised manuscript.

5)The description in Line 51-52 is not correct. Anionic hybrid copolymerization is quite different from the anionic ring-opening polymerization in mechanism, the former is absolutely not the expansion of the latter. The sentence is suggested to correct as follows:

The anionic ring-opening polymerization of S₈ was first reported by Penczek et al. in 1978. Very recently, Yang et al. discovered the anionic hybrid copolymerization of S₈ with acrylate. We agree with the point that the anionic hybrid copolymerization should be distinctive from the anionic ring-opening polymerization. The relative description has been accordingly corrected, following your suggestion, in the revised manuscript.

Reviewer #3 (Remarks to the Author):

Qu, Zhang and coworkers have presented a clever use of strained dithiolanes to recruit otherwise wasted sulfur to manufacture new materials. The synthesis is green, free of catalyst or solvent. The wide scope of monomer lends the materials a wide range of accessible property parameters. As a demonstration, the polymer is shown to be elastic, self-healable, adhesive, and degradable. I strongly support the publication of this manuscript. I have some technical questions that I hope the authors can offer to clarify:

We appreciate the positive and comments and constructive suggestions of this referee. Additional experiments have been performed to address the issues mentioned, and new data have been added into the revised manuscript and SI.

1) Page 3 Line 62 / Figure 1. The authors proposed a radical pathway. Is it possible that S–S can undergo direct exchange with S₈? In other words, if radical quencher is added / present, will the polymerization still be robust? I ask this question, because I imagine the reaction would have a broader impact or utility if the reaction is tolerant with impurities.

Accordingly, we have performed several additional experiments to test the effect of radical quenchers/scavengers in our copolymerization system. Commonly used radical trapping agents, e.g., TEMPO, polymerization inhibitor (p-Toluidine, chloranil, and hydroquinone), sulfur radical quencher (5-hexen-1-ol), have been tested, and the resulting solids were characterized by XRD and Raman spectroscopy (Fig. S28–29). XRD data revealed that the resulting copolymers with TEMPO, P-toluidine, or chloranil are amorphous. In the case of 5-hexen-1-ol or hydroquinone, a small amount of sulfur precipitation (23°) was observed. Raman spectra also suggest the occurrence of copolymerization. These results support the robustness of our system and its tolerance towards impurities, which should be attributed to the advantage of “in-situ” and continuous generation of sulfur radicals from cyclic disulfide monomers at the used temperature.

Supplementary Fig. 28 | Additional experiments to test the effect of radical quenchers/scavengers in the copolymerization system.

Supplementary Fig. 29 | XRD and Raman analysis. (A) XRD patterns and (B) Raman spectra of poly(S_8 /TAA=1/1) with 0.1 eq various radical trapping agents, polymerization inhibitors, or sulfur radical quencher.

2) Page 3 Line 65, I guess the use of “network” is not justified? And the following sentence seemed a bit confusing. Did the author intend to say the reactions is promoted by ROP? The referee might refer to the network of chemical reactions. Here the term of “network” is specific to the architecture of the resulting copolymers coined as supramolecular/noncovalently crosslinked network. We have changed the mentioned terms in case of potential misunderstanding in the revised manuscript.

3) Figure 2c. There are some unidentified peaks in the polymer sample, e.g., ~1.8 ppm. The reaction between S_8 and TAA is supposed to be clean and the starting materials looks clean, too. Then, can the author comment on what are the other species present in the polymer sample? This is important, since the authors claimed the free of need to purify. The ¹H peaks of the polymer sample have been reidentified. The peak at ~1.8 ppm belongs to proton b' instead of impurities. We don't deny that this radical and dynamic reaction is ideally pure. For example, the monomer conversion is estimated at 90%. Also, the reddish color of the resulting samples suggests the existence of some oligomeric S_x species in the bulk solids. However, we claim the free of the need to purify because the minor existence of monomers didn't affect the material performance profoundly, which has been shown by our additional experiments (Fig. S37). All these discussions and new data have been added to the revised manuscript and SI.

Fig. 2 | Structure characterization of poly(S_8 /TAA) copolymers. c, 1H spectra of poly(S_8 /TAA=1/2) and TAA monomer in DMSO- d_6 .

Supplementary Fig. 37 | Purification with CS_2 . (A) The images of the purification process. (B) 1H NMR spectrum, (C) Solid-state UV-Vis spectra, (D) TGA analysis, (E) DSC analysis, and (F) ATR analysis of the poly(S_8 /TAA=1/2) after purification.

4) Figure 2d, top panel. The boundary between the green and blue regions seem arbitrary? I wonder if some reference spectrum or its assignment should be provided in the SI. Also, I don't feel like broadening is the right word for C-S vibration, it simply nearly disappeared

in the polymer. Stating that it is broadened appears to be a cyclic argument to me: that you assume it is the compound, and therefore it should be broadened?

The Raman spectra included the copolymer sample and two reference samples (TAA and S₈), together compared with the literature's value of similar poly(sulfur) materials (e.g., *J. Am. Chem. Soc.* **2023**, 145, 14539–14547; *Angew. Chem., Int. Ed.* **2022**, 61, e202115950; *Macromolecules* **2010**, 43, 4133–4139). We agree that it is a good idea to reassign the vibration bands with combined species because the whole region relates to the S–S–S and S–S bonds as a signature of successful copolymerization. Thus, the assignment and discussion have been properly adapted in the revised manuscript.

5) Page 7 Line 116: These are different in terms of bond connectivity – feels like a strange statement to me. I assume the authors meant the spectra is different because of changes in connectivity?

The term has been rephrased in the revised manuscript.

6) Page 7 Line 119 / Fig S22-S24: The MALDI showed different patterns in S_x segment. Does that mean different monomers led to polymers of various multi-S segments? Then how does the XPS corroborate with the length of S in the polymer, and how about different S_x / monomer ratio? How about mechanical and other properties shown in this manuscript? Is a long S_x better or worse, for example. I felt this is probably the most important issue that is kind of weak in this paper. I hope the material properties can be better correlated with the structures.

Yes, the S_x segment is of various configurations in the current copolymer system because the disulfide-rich backbone is dynamic and thus, reconfigurable. In the current system, we don't aim at precisely controlling the multi-S segment distribution. However, we still can manipulate the global stoichiometric ratio of the copolymer by varying the feedstock ratios. Thus, we can establish a qualitative relationship between the S_x ratio and the mechanical performances of the copolymers. To strengthen the discussion and scientific insights regarding this point, we analyzed and compared mechanical tensile curves within the samples with different sulfur ratios (Fig. S38). For example, The Young's modulus of poly(S₈/TAA) was increased from 68 MPa (S₈ : TAA = 1 : 5) to 215 MPa (S₈ : TAA = 1 : 1), while the Young's modulus of poly(S₈/TABA) was increased from 4.2 MPa (S₈ : TABA = 1 : 3) to 9.1 MPa (S₈ : TABA = 1 : 1). This feature of mainchain-structure-dependent mechanical properties enables tunability of the resulting copolymers by controlling the copolymerization ratios of the two monomers, i.e., cyclic disulfides and elementary sulfur. The discussion has been added to the revised manuscript.

Supplementary Fig. 38 | Stress-strain curves of copolymers. (A-D) Stress-strain curves of the various copolymers. The measurement was carried out with a strain rate of 50 mm/min at RT.

7) Page 9 Line 165: “Systematic investigations of the copolymer samples with different ratios revealed mechanical tunability of the materials” – I think the authors can be more quantitative here since they have the numbers, include the sentence that follows. What is the range for the state-of-the-arts?

We have added a more quantitative comparison of the mechanical properties with different ratios in the revised manuscript. The state-of-the-art (from 2013 to 2023) regarding the poly(sulfur) copolymers has been summarized and presented in Supplementary Table S2, showing that the mechanical properties of our copolymer materials do stand out.

8) Page 13, Line 212 / Figure 4h: What is the role of DMSO and how does it degrade the polymer? What about other polar solvent such as DMF and HMPA?

The copolymer is dissolvable in DMSO. After checking with UV-Vis spectra, we conclude that this dissolution process does not involve depolymerization due to the absence of characteristic absorption bands of cyclic disulfides. The corresponding sentence has been corrected in the revised manuscript. For DMF and HMPA, the copolymer is also soluble but not degradable.

Supplementary Fig. 54 | UV-vis spectra of poly($S_8/TAA=1/2$) and poly($S_8/TAH=1/2$) in DMF, DMSO, or Hexamethylphosphoramide (HMPA) at room temperature.

REVIEWERS' COMMENTS

Reviewer #1 (Remarks to the Author):

Revision is suitable for publication.

Reviewer #2 (Remarks to the Author):

The manuscript entitled "Converting inorganic sulfur into degradable thermoplastics and adhesives by copolymerization with cyclic disulfides" was well revised with my concerns addressed. I think it could be published in Nature Communications.

Reviewer #3 (Remarks to the Author):

I thank the authors for their careful and thorough responses to my inquiries. The manuscript is now in a much better shape and clarity for publication. I have no further questions.